# Monitoring Drought through the Lens of Landsat: Drying of Rivers during the California Droughts

**Shang Gao** [1], **Zhi Li** [1], **Mengye Chen** [1], **Daniel Allen** [2], **Thomas Neeson** [3] and **Yang Hong** [1,*]

1   Hydrometeorology and Remote Sensing Laboratory, School of Civil Engineering and Environmental Science, University of Oklahoma, Norman, OK 73019, USA; shang.gao@ou.edu (S.G.); li1995@ou.edu (Z.L.); mchen15@ou.edu (M.C.)
2   Department of Biology, University of Oklahoma, Norman, OK 73019, USA; dcallen@ou.edu
3   Department of Geography & Environmental Sustainability, University of Oklahoma, Norman, OK 73019, USA; neeson@ou.edu
*   Correspondence: yanghong@ou.edu; Tel.: +1-405-996-8128

**Abstract:** Water scarcity during severe droughts has profound hydrological and ecological impacts on rivers. However, the drying dynamics of river surface extent during droughts remains largely understudied. Satellite remote sensing enables surveys and analyses of rivers at fine spatial resolution by providing an alternative to in-situ observations. This study investigates the seasonal drying dynamics of river extent in California where severe droughts have been occurring more frequently in recent decades. Our methods combine the use of Landsat-based Global Surface Water (GSW) and global river bankful width databases. As an indirect comparison, we examine the monthly fractional river extent (FrcSA) in 2071 river reaches and its correlation with streamflow at co-located USGS gauges. We place the extreme 2012–2015 drought into a broader context of multi-decadal river extent history and illustrate the extraordinary change between during- and post-drought periods. In addition to river extent dynamics, we perform statistical analyses to relate FrcSA with the hydroclimatic variables obtained from the National Land Data Assimilation System (NLDAS) model simulation. Results show that Landsat provides consistent observation over 90% of area in rivers from March to October and is suitable for monitoring seasonal river drying in California. FrcSA reaches fair (>0.5) correlation with streamflow except for dry and mountainous areas. During the 2012–2015 drought, 332 river reaches experienced their lowest annual mean FrcSA in the 34 years of Landsat history. At a monthly scale, FrcSA is better correlated with soil water in more humid areas. At a yearly scale, summer mean FrcSA is increasingly sensitive to winter precipitation in a drier climate; and the elasticity is also reduced with deeper ground water table. Overall, our study demonstrates the detectability of Landsat on the river surface extent in an arid region with complex terrain. River extent in catchments of deficient water storage is likely subject to higher percent drop in a future climate with longer, more frequent droughts.

**Keywords:** drought; Landsat; river surface extent; seasonal drying; NLDAS

## 1. Introduction

Drought is an irregular period of low water abundance or water scarcity [1] when water availability cannot fully meet the needs for agriculture, urban supplies, and ecosystems. The monitoring and mitigation of drought largely target meteorological drought, which is defined by the lack of precipitation and surplus of evapotranspiration, as well as the agricultural drought occurring as a consequence of this deficit [2]. Hydrological drought featuring below-normal water discharge develops over time from meteorological drought and is generally more varied in space and time because of the heterogenous antecedent conditions, e.g., soil moisture and surface- and ground-water storage [3]. This is partially reflected by the complex drying patterns of rivers and defines the thresholds and boundaries of river ecosystem health at a local scale [4]. Monitoring the spatiotemporal

dynamics of hydrological droughts is thus essential for better assessing and understanding the impacts on ecosystems and human societies.

Surface water has been impacted by hydrological droughts in a wide variety of climates worldwide, ranging from equatorial [5], cool temperate [6], to semi-arid [7]. This has become especially relevant in the western United States, where frequent droughts have historically shaped water resource management in the region. In California, droughts have become more common and severe in the recent decades and are projected to be more frequent in a future warming climate [8]. For instance, the 2012–2015 drought caused serious impacts to human life and ecosystems. The four years saw below-average precipitation consistently and almost the whole state was in 'Extreme' or 'Severe' drought in 2015; streamflow was also below average at most gauged streams [9].

The traditional monitoring system of river dynamics relies on the in-situ observations from networks of stream gauges/loggers and field surveys, which are limited by the logistical challenges, spatial density, and temporal frequency of the collected data [10–13]. In recent decades, remote sensing, as an emerging technology, has come to provide information on surface water over large spatial extent at a high temporal resolution. Tremendous research effort has been invested in the remote sensing of surface water, as reviewed in detail by Huang et al. [14]. In general, automated segmentation of water bodies based on satellite imagery is classified into rule-based [15–18] and machine/deep learning approaches [19–21]. The maps of segmented water bodies in turn enable studies on surface water dynamics analyzing changes in space and time [22–25]. The state-of-the-art research by Pekel et al. [26] has produced the Global Surface Water (GSW) dataset by using 3 million Landsat scenes, which essentially indicates whether surface water was present at any location of interest from March 1984 to December 2018 on a monthly basis. GSW also includes products related to water changes, such as water occurrence, recurrence, seasonality, and change intensity.

In terms of river water particularly, the remotely-sensed surface water estimates, in conjunction with advances in high performance and cloud-based computing, have been converted to several large-scale datasets of river geomorphology and improved understanding of river dynamics. Yamazaki et al. [27] developed the Global Width Database for Large Rivers using Shuttle Radar Topography Mission and Water Body Data for large rivers (width > 300 m). A more refined product, Global River Widths from Landsat, was produced by Allen and Pavelsky [28] for rivers wider than 30 m. Built upon these global river width estimates, Lin et al. later [29] produced a global bankful width dataset using machine learning techniques. Yang et al. also [30] produced the Multi-temporal China River Width (MCRW) dataset, the first 30-m multi-temporal river width dataset for China during 1990–2015, which includes estimates under both seasonal fluctuations and dynamic inundation frequencies.

With these datasets, new opportunities have emerged for better characterization and interpretation of the spatiotemporal patterns of surface water in rivers at the local, regional, and global scales. As demonstrated by Allen et al. [31], the current satellite-imagery archive is on average representative of the flow regimes present along Earth's large rivers and is valuable for analyzing their long-term behaviors. Additionally, much previous research on California droughts has focused on its severity [32–34] through meteorological, hydrological, agricultural, and socio-economic indices [35,36]. Researchers also investigated the causes of droughts, e.g., climatic variability [37] and human drivers of climate change [38–40]. However, fewer studies have investigated its hydrological or ecological consequences on rivers, or the post-drought recovery of river water [9,41]. In this regard, our goal is to characterize and interpret the seasonal drying dynamics of river extent as detected the by Landsat datasets, during dry periods of California droughts and the immediately following wet periods, with the extreme 2012–2015 drought serving as a typical case. We address the following research questions: (1) How is the detectability of Landsat on river extent dynamics in an arid region such as California? (2) What are the characteristics of the rivers during- and post- the extreme 2012–to–2015 drought? (3) How

can the observed dynamics of river extent be attributed to hydroclimatic and geographical factors? The paper is organized as follows. Section 2 describes the study area, data, and methods used in this study. Section 3 presents the results of statistical analyses. Section 4 discusses limitation and in-depth implication of the study. Section 5 concludes the study and proposes future directions.

## 2. Study Area, Data, and Methods

We conduct our analysis upon rivers in the seven climate divisions of California (Figure 1A). The river reaches in this study are part of the global bankful width dataset [29] for all reaches detectable by Landsat (bankful width >= 30 m). Figure 1B shows the 2071 reaches in the study area with corresponding bankful widths. Note that this study also includes the Colorado river which is located at the boundary of California and Arizona. The rivers in the central valley are dominated by groundwater with shallow water depths (Figure 1C). The rivers in the central and southern coastal areas and Sierra Nevada are subject to regulation, whereas those in northern coast and the northeast interior basins are more naturalized (Figure 1D).

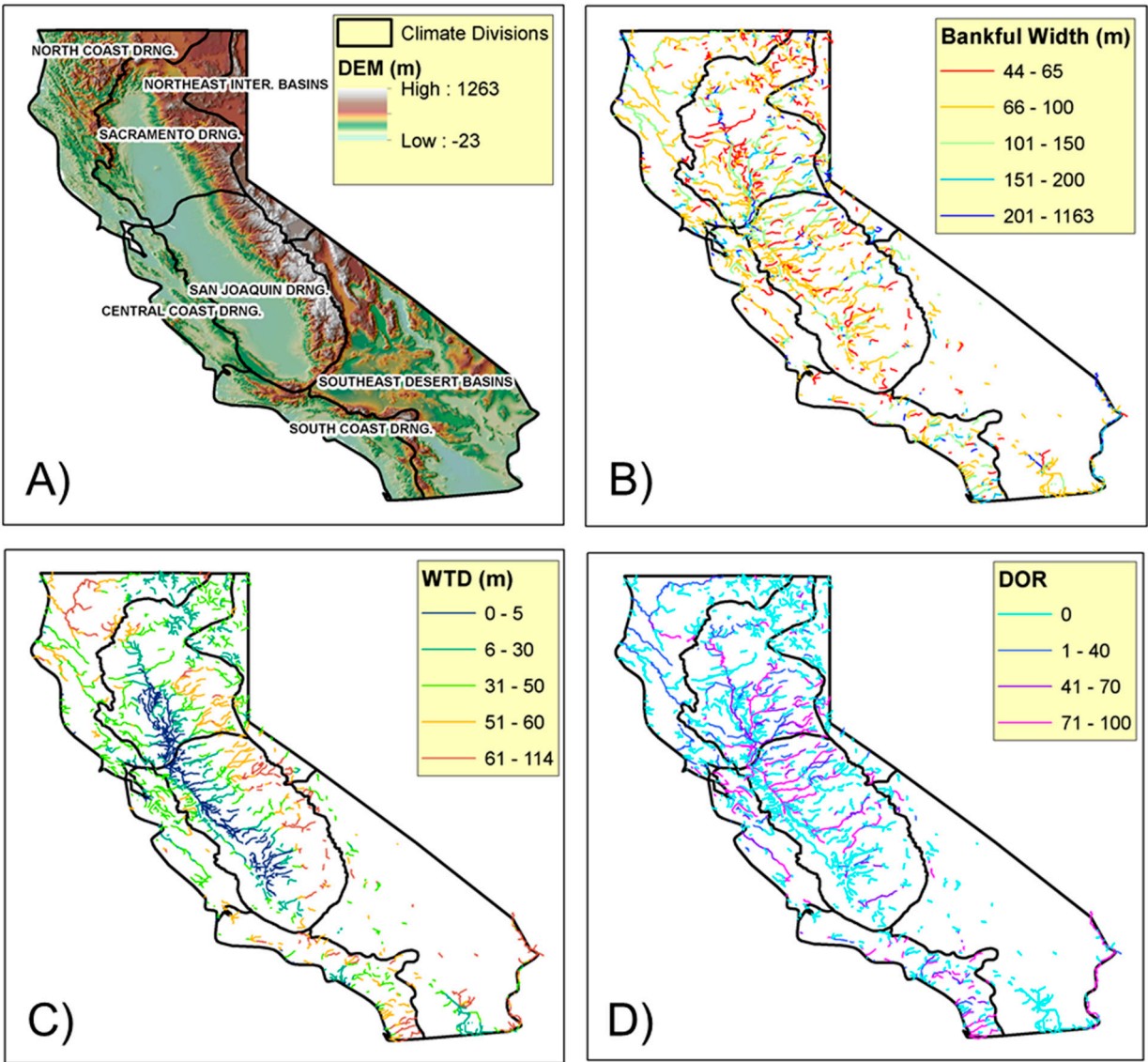

**Figure 1.** Maps of (**A**) study area, seven climate divisions, (**B**) bankful width, (**C**) water table depth (WTD), and (**D**) degree of regulation (DOR) of the 2071 river reaches detectable by Landsat. River attributes are retrieved from Lin et al. [29].

### 2.1. Landsat Datasets and River Extent Metrics

We target the surface water extent in rivers in this study. Our methodology starts with defining an examination extent for river water, which should encompass the full range of variability due to various factors, e.g., shifts of river channels over time and channel widening (narrowing) during floods (droughts). We utilize the monthly history product of Global Surface Water (GSW) dataset [26] which consists of three values, i.e., water, no water, and no observation at 30-m resolution. Within a 34-year period from January of 1985 to December of 2017, the union of all 'water' pixels in GSW monthly history is taken to mask all locations ever detected as surface water, referred to as '*max_extent*' hereafter. Figure 2A illustrate *max_extent* at a river confluence near 38°N and 121.5°W which masks not only river extent but also adjacent non-river water bodies, e.g., ponds, pools, and potholes. To focus only on rivers, we use the river centerline of the global river bankful width dataset [29] to extract river extent from *max_extent* based on connectivity to the river centerline, meaning isolated water bodies are excluded (Figure 2B). A large search distance of 15,000 m is used to examine connectivity to river centerlines and ensure the full widths of large rivers or anabranching/braided rivers are covered. Within this finalized maximum river extent, we develop a monthly time series of fractional river extent (FrcSA hereafter), defined as the ratio of the number of water pixels to the number of observed pixels, i.e., sum of water and no-water pixels (Figure 2C). This calculation is made within the catchment (local drainage area) for each of the 2071 river reaches. FrcSA is estimated instead of the actual water-covered area because the Landsat observations are spatiotemporally discontinuous and also were improved over the past 35 years [31]. In addition, the FrcSA monthly series is filtered to exclude cases of excessive no-observation pixels (over 10% of the total pixels). Such filtration thus produces gaps in the FrcSA time series, which is illustrated later in Section 3.1.

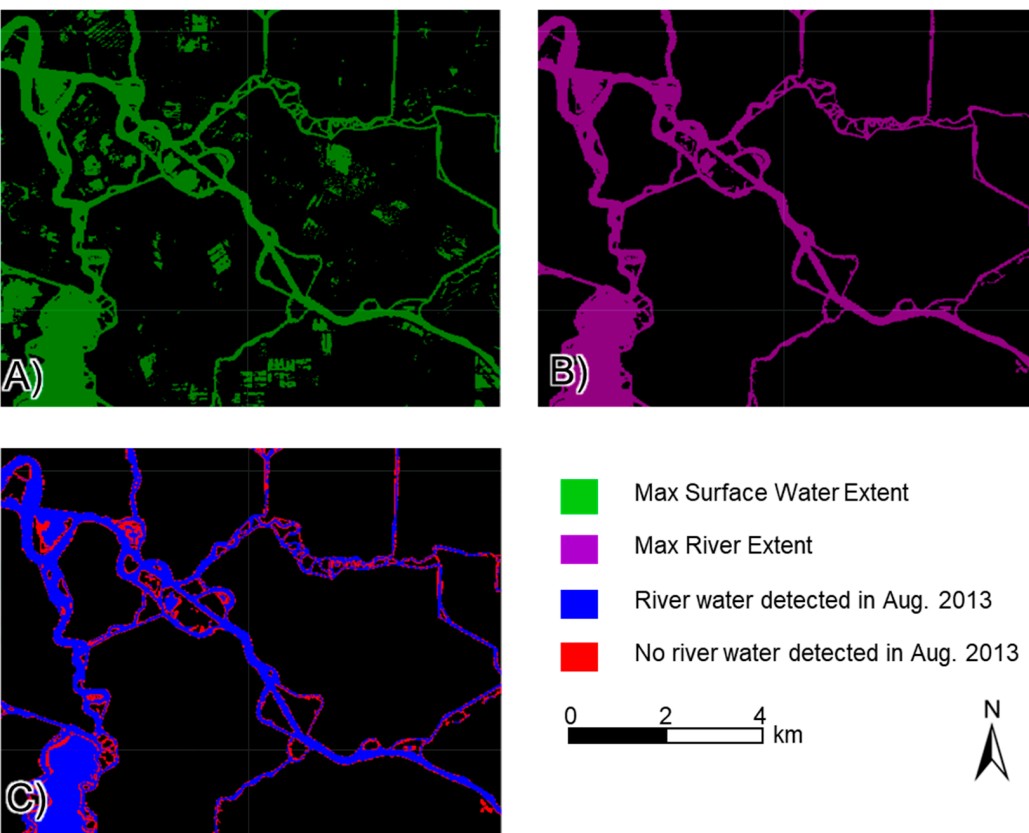

**Figure 2.** Examples of (**A**) maximum surface water extent (green) and (**B**) the maximum river extent (purple) extracted based on connectivity to river centerline, and (**C**) the water and no-water pixels in the GSW monthly water history product masked by the maximum river extent in August of 2005. The location is near 38°N and 121.5°W.

With FrcSA indicating monthly river extent of individual reaches, we also examine the average surface water presence within some defined time period. At a given 30-m pixel, water occurrence (WO) is defined as below:

$$WO = \sum WD / \sum VO \tag{1}$$

where WD is water detection and VO is valid observation which includes WD and no-water detection. WO can reveal the perennial (WO = 1), intermittent (0 < WO < 1), and dry (WO = 0) portions of individual reaches during the defined time period.

In order to differentiate river behaviors during drought and non-drought months (years), we utilize the Palmer Drought Severity Index (PDSI) for the same seven climate divisions as in Figure 1A from the National Climatic Data Center [42]. We identify droughts in same fashion as Dettinger et al. [43] where drought endings are when PDSI cross a threshold value (PDSI = −2) while drought starts are when PDSI cross (from wetter to drier conditions) and stay below the same threshold for at least six months.

As an indirect comparison, we examine the correlation of FrcSA with streamflow from USGS gauges. Candidate stream gauges are selected from USGS's Geospatial Attributes of Gages for Evaluating Streamflow, version II reference database [44,45]. A data filtering ends up with 187 gauges that can be co-located with reaches and consists of less than 10% of missing data over the 34-year period from 1 January 1985 to 31 December 2017. The daily streamflow values are temporally averaged to monthly time series for the correlation analysis.

## 2.2. Attrubtion of Surface River Extent Dynamics

### 2.2.1. Hydro-Climatic Data and Related Metrics

We utilize the North American Land Data Assimilation System (NLDAS) which is a land surface model that outputs land surface states and fluxes using parameters obtained/derived from satellite observations, ground observations, and (re)analyses [46]. In particular, we extract rainfall, snow water equivalent, and root-zone soil moisture from the Noah version 2.1 monthly outputs during the same period as the Landsat observation (January 1985 to December 2017). The original 1/8° data are resampled to 1/160° using bilinear interpolation to improve resolution near catchment edges. Areal averages are then calculated over the contributing area towards target reaches. The hydrography information for the global bankful width data [47], including flowlines and catchment polygons, are used to identify the contributing areas. For this analysis, we select independent reaches/catchments of Strahler order one or two to achieve sufficient data samples (more than high-order reaches). As a result, this analysis consists of 863 reaches, i.e., about 40% of the total 2071. Root zone soil moisture is considered an indicator for water storage and its monthly times series is computed for correlation analysis with FrcSA. We then compute annual time series of cumulative winter (November to March) precipitation (Pw). We also calculate the humidity index (P/PET) which is ratio of cumulative annual precipitation to cumulative annual potential evapotranspiration (PET). The humidity index is computed for each water year (October to the following September) and then averaged to produce one long-term mean P/PET for each reach/catchment.

### 2.2.2. Correlation and Elasticity Analysis

This part of the analysis is conducted to determine the natural causal factors/indicators for the dynamics of river extent. Certain terrestrial water storage components (e.g., soil water or ground water) can exhibit a storage-discharge relationship with FrcSA, which is manifested as positive correlations at some temporal lags. Therefore, we examine the lagged correlation between monthly root zone soil moisture and monthly FrcSA with the former preceding the latter. Lags of 0 to 3 months are attempted and the one generating the highest correlation coefficient is regarded optimal.

Elasticity analysis quantifies the climatic sensitivity of FrcSA to some forcing variables. In this study, the FrcSA elasticity is defined as the proportional change in FrcSA for

a fractional change in winter precipitation (Pw). For each year at each catchment, we normalize the annual mean Pw and annual mean FrcSA (average of monthly FrcSA from March to October) by their average for all years. At all catchments, normalized annual mean FrcSA is fitted to a linear function of normalized annual mean Pw. The slopes of these linear relationships estimate the percentage change in FrcSA in response to a unit percentage change in Pw.

## 3. Results

### 3.1. Dectectability of Landsat on River Extent Dynamics

Figure 3 shows the probability distribution of Landsat observation rate for the 2071 reaches, i.e., the fraction of observed pixels, in each month. It can be found that majority of reaches are well observed from April to October with the median observation rate above 0.9. However, the reaches are poorly represented by Landsat dataset during November, December, January, and February. Landsat is thus considered a good source for monitoring streams in only the summer and fall seasons, in terms of observation rate. In California, the precipitation is comprised mainly by rainfall, except for the Sierra Nevada where snowfall can reach up to 90% of the annual precipitation. The rain-dominant catchments feature peaks of both rainfall and streamflow from December to the following February, whereas, in snow-dominant catchments, streamflow peaks occur in summer months (May to July) due to snow melting [48]. Therefore, the rewetting of rivers is barely detected by Landsat in the majority of rain-dominant California during winter months. The increased no-observation pixels in winter could be due to the cloud shadow in the wet season.

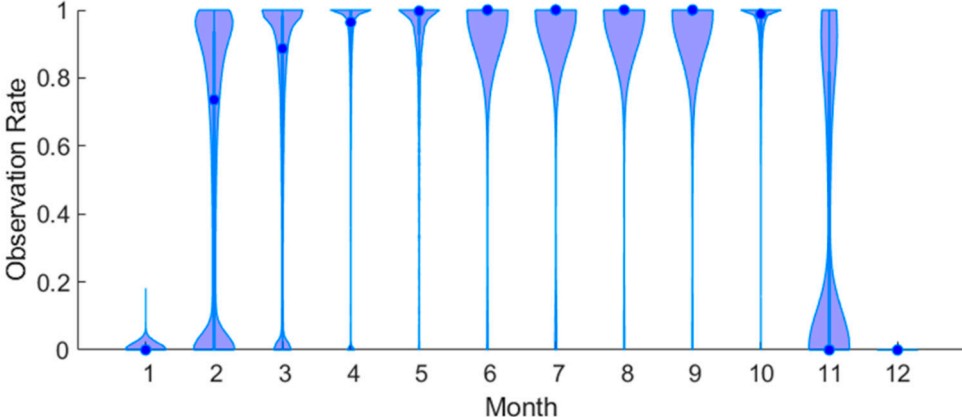

**Figure 3.** Violin plots (probability distribution) of observation rates for reaches in each month. Observation rate is defined as the fraction of observed pixels within maximum river extent as illustrated in Figure 2B. Filled dots represent the median values.

As an indirect comparison, the correlation of monthly streamflow and FrcSA is examined in 187 reaches with co-located USGS gauges. Out of 121 reaches with statistically significant results ($p$ value < 0.05), nearly half (54) result in fair (above 0.5) correlation coefficients with USGS gauges (Figure 4A). The north coast division and the Sacramento division see more co-located gauges with higher correlation values with the other divisions. Figure 4B,C show the USGS streamflow and FrcSA for the reaches with the best and worst correlations, respectively. The evident discontinuity seen in the FrcSA time series is due to a threshold observation rate of 0.9 used to exclude some months when no observation pixels bring uncertainty to FrcSA estimates. Although both are at monthly time steps, the two compared time series are computed differently: FrcSA is based on Landsat imagery captured at instantaneous moments within a given month while monthly USGS streamflow is temporal average of daily values. This distinction in the sampling frequency of raw data can reduce the correlation between the two. Further, the lower correlations in mountainous reaches in the North Sierra Nevada are likely associated with the canopy blockage over the

rivers. Drier climate divisions in the south feature narrower rivers with smaller streamflow, which could exceed the detectability (30-m resolution) of the Landsat. In addition, the co-located USGS gauges are of a small fraction of all 2071 reaches in California, indicating potential of Landsat as a supplemental monitoring source for rivers.

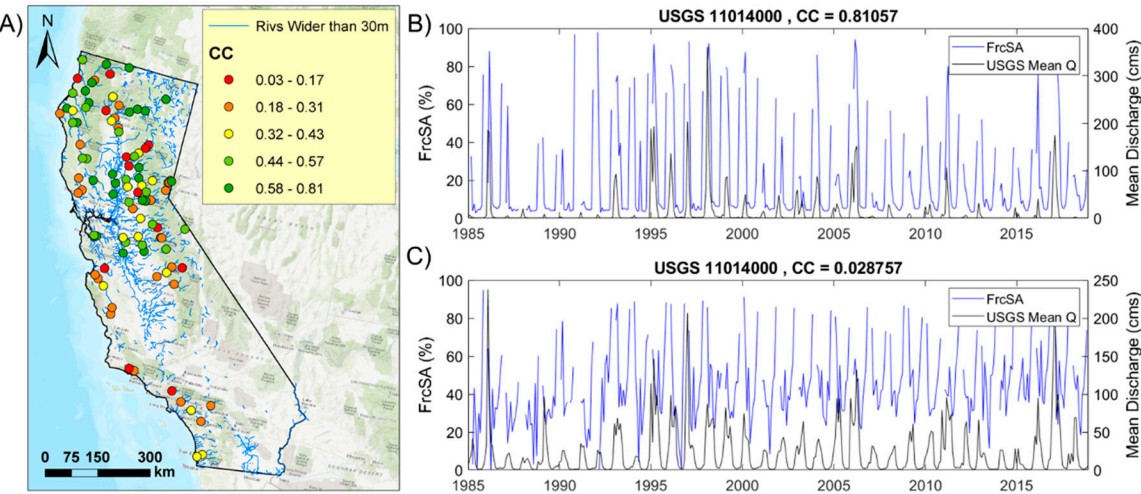

**Figure 4.** Correlation coefficients of monthly FrcSA and monthly mean streamflow at co-located 187 USGS gauges (**A**), the time series at the gauges with the highest (**B**) and lowest (**C**) correlations.

We define drought and non-drought periods based on PDSI for the seven climate divisions (see Figure S1 in supplementary material). Two historical droughts, starting in 2006 and 2012, respectively, can be identified across all the climatic divisions. Considering both mean PDSI and the duration (Table S1 in supplementary material), the 2012–2015 drought is the most severe one during the Landsat period (1984 to present). Given the sharp contrast of PDSI during drought and non-drought periods, monthly water occurrence (WO) in both periods is also examined for the seven climate divisions (Figure 5). For the calculation of WO in a given climate division, we take the ratio of all water detections to valid observations in rivers of that area (Equation (1)). All seven climate divisions see drying of streams (shrinking of river extent) in summer, with WO dropping more (~30%) in northern divisions (north coast, northeast interior basins, Sacramento) than in the other southern, drier climate divisions (~10%). Additionally, the river extent in drought years is almost consistently lower throughout summer and fall than in non-drought years.

One signature trait for California droughts is the slow, gradual decay of PDSI at drought starts along with abrupt endings often triggered by atmospheric river events and indicated by a PDSI jump [43] (Figure S1). Therefore, significant changes of river extents can be expected before and after the drought breaks (endings). Taking the 2012–2015 drought as an example, we further examine the contrast of river extent and water occurrence in rivers before and after the drought ending. Figure 6 shows the change of dry, intermittent, and perennial areas in all reaches in year 2017 compared to year 2014. These two years are chosen to respectively represent during- and post-drought conditions as the drought had ended by year 2015 or year 2016 for all climatic divisions. The one-year WO is first calculated for all pixels in rivers and then classified into dry, intermittent, and perennial cases corresponding to WO = 0, 0 < WO < 1, and WO = 1, respectively. After the drought ended, 43% and 58% of the 2071 reaches had increased perennial and intermittent areas (blue-colored reaches in Figure 6A,B), respectively; whereas decreased dry areas were seen in 65% of the reaches (red-colored reaches in Figure 6C). The large fractions of reaches experiencing changes after the drought ending indicate drastic switches of flow regimes in the whole of California. To further put the 2012–2015 drought into historical context, we calculate the annual mean FrcSA (average of FrcSA from March to October) for all 34 years from 1985 to 2018. A total of 1449 reaches result in FrcSA time series with consistently good

(>0.9) observation rates. For each reach, the annual mean FrcSA are ranked from high to low and Figure 7 shows the ranks during the four years from 2013 to 2016. During the drought, river extent in numerous rivers decreased, reached historical lows in 2014 and 2015, and recovered in 2016. Forty seven percent (47%) of the examined reaches ranked above 30th, i.e., among the top 4 driest years in Landsat record, at some point during the drought, and 332 reaches reached the very driest (34th) in Landsat history. Across the drought ending, the reaches experienced an average rank increase of 14.3, i.e., rank difference between year 2016 and the driest among years 2013, 2014, and 2015.

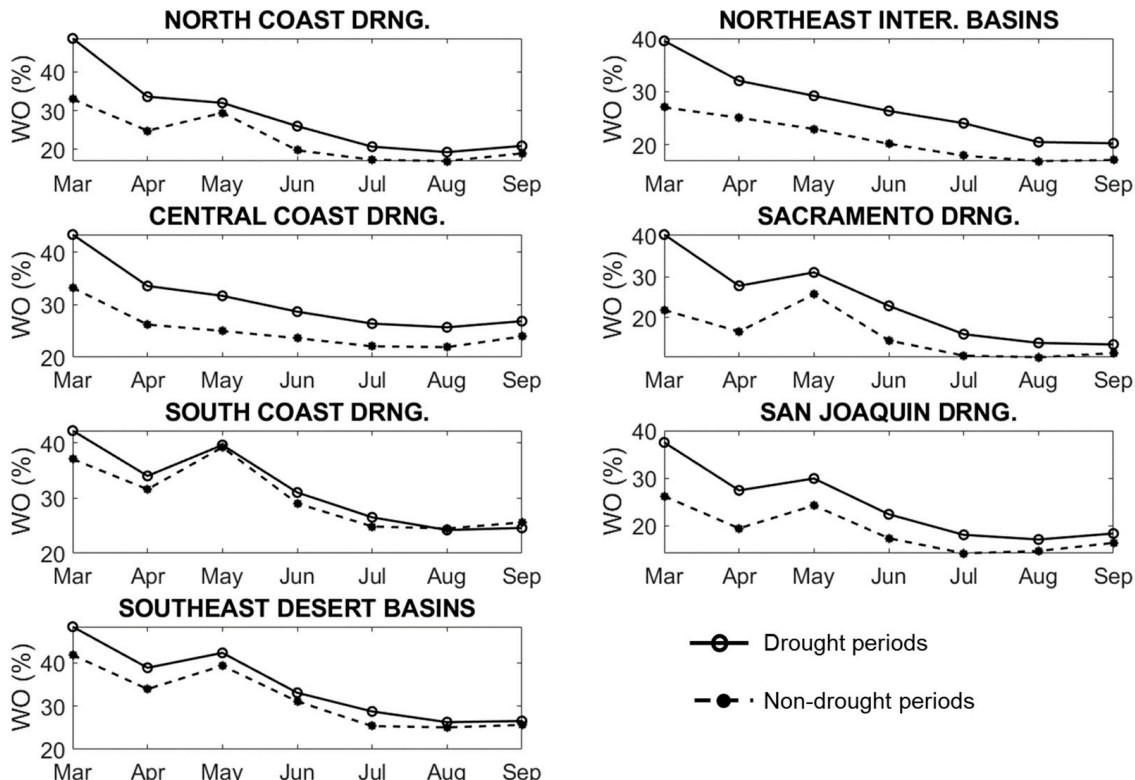

**Figure 5.** Mean river extent/area change during seasonal drying from March to September averaged over the drought and non-drought periods.

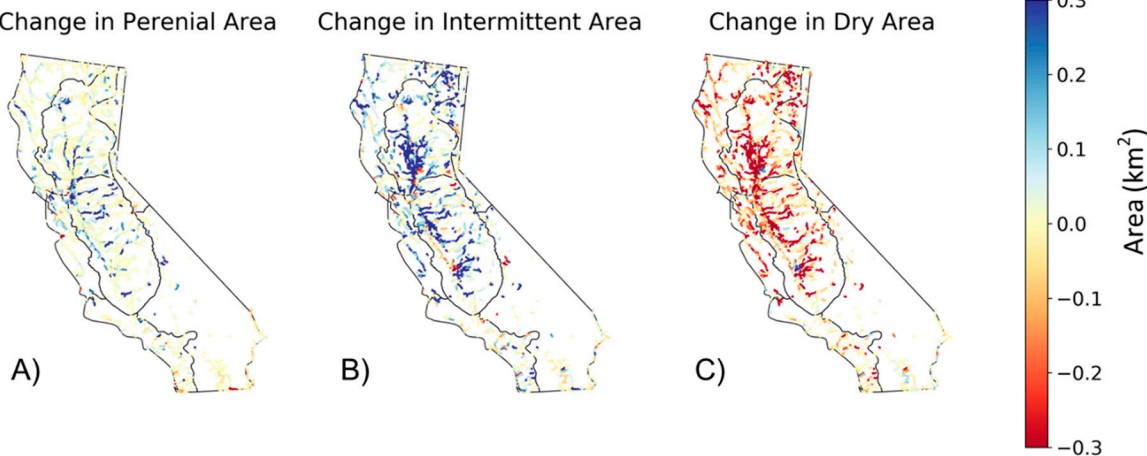

**Figure 6.** (**A**) Change of perennial, (**B**) intermittent, and (**C**) dry area of 2017 river reaches in post-drought year 2017 com-pared to during-drought year 2014.

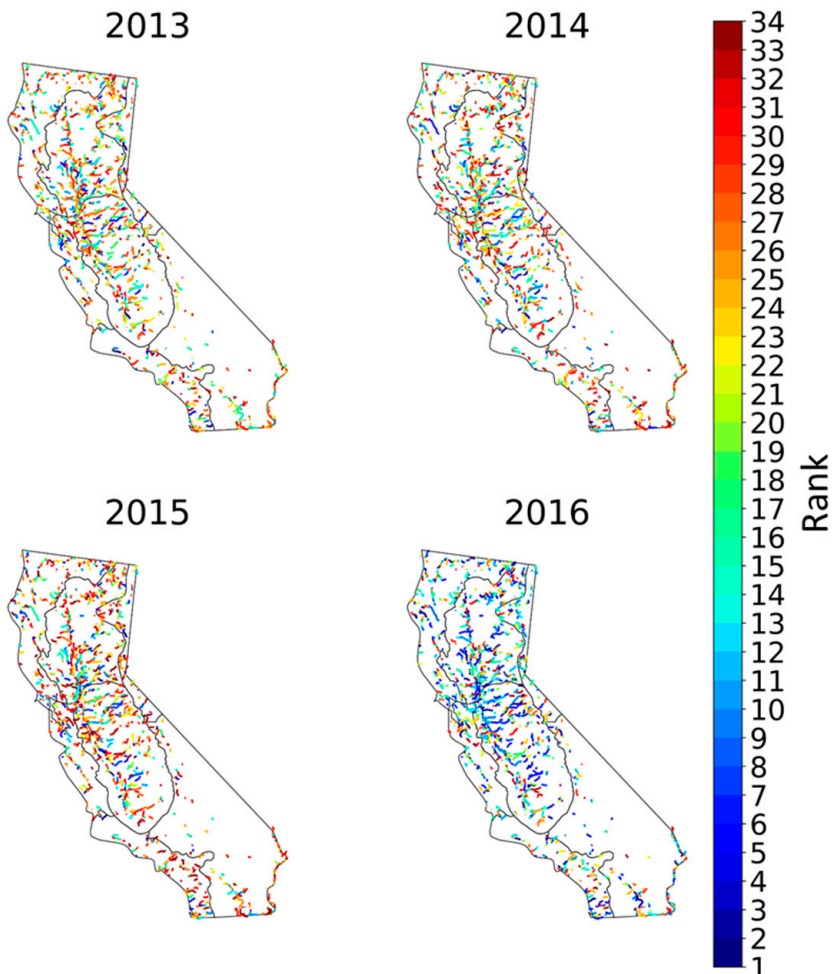

**Figure 7.** Ranks of annual mean FrcSA from high to low among 34 years of Landsat observation for 2071 river reaches in year 2014 to 2016. Ranks of 1 and 34 mean highest and lowest FrcSA during the 34 years, respectively.

In addition, the 2012–2015 drought is compared with the 2007–2009 drought which is also identifiable in all the seven climate divisions based on PDSI (Figure S1 and Table S1). We examine the annual mean normalized FrcSA, i.e., average of FrcSA from March to September divided by long-term (34-year) mean, during the one year immediately preceding and following the drought endings (Table S1), as shown in Figure 8. Comparing the two droughts, the 2012–2015 drought is more severe, causing the during-drought FracSA values to be lower than those of the 2007–2009 drought in all the climate divisions. Furthermore, the extraordinary drought-ending wet events in 2016 leads to a strong recovery of water, causing the majority of rivers to exceed long-term mean FrcSA in five of the seven climate divisions; in comparison, the change of FrcSA post the 2007–2009 drought is less significant. Interestingly, the FrcSA in the northeast interior basins is insensitive across both drought endings despite a >3 PDSI increase. Overall, the Landsat observations can clearly depict the dynamics of rivers in response to the two extremes of regional climatology during and after the 2012–2015 drought.

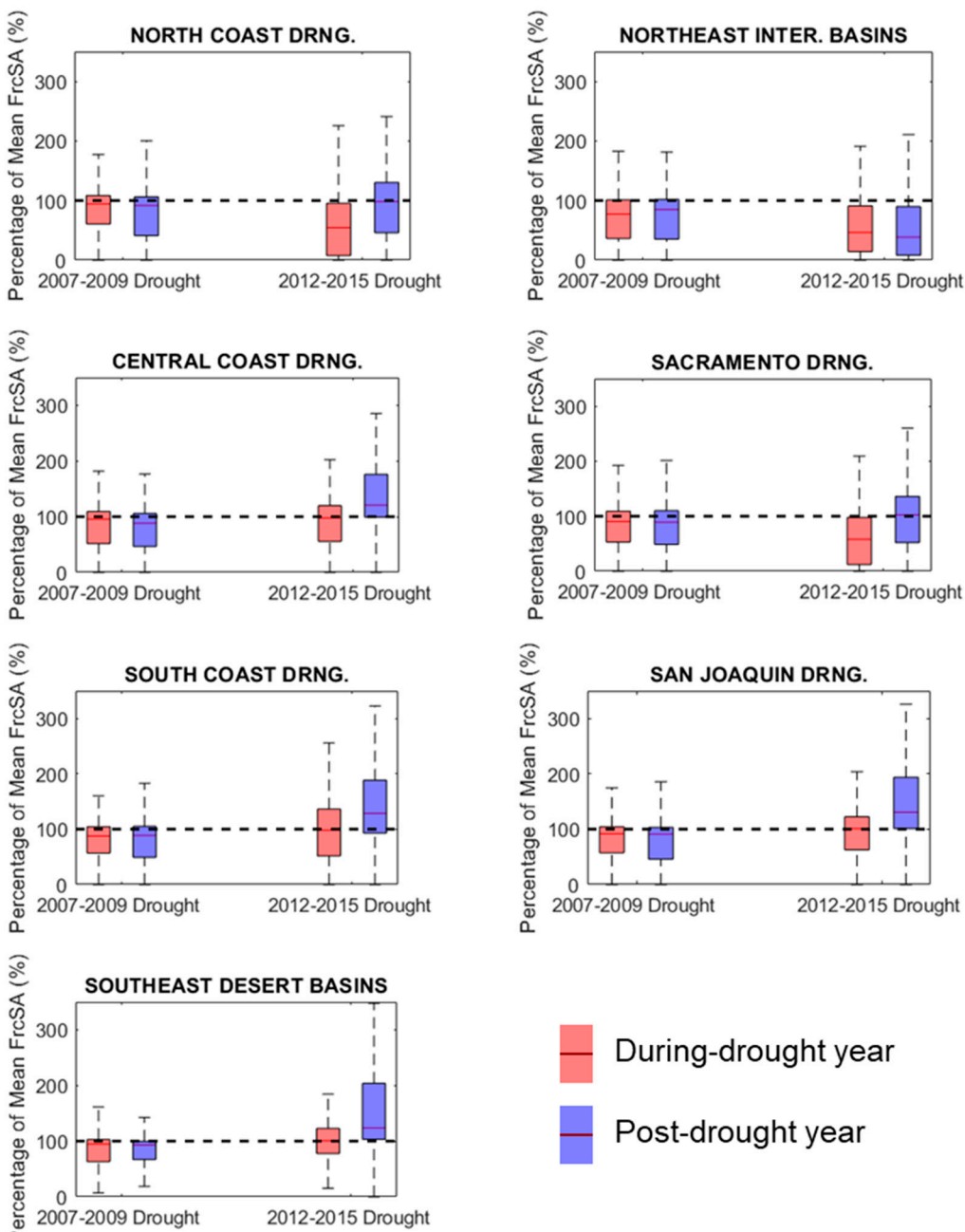

**Figure 8.** Normalized FrcSA (by long-term mean) of reaches in one year during droughts and one year after droughts for the 2007–2009 and 2012–2015 droughts and the seven climate divisions.

### 3.2. Results of Correlation and Elasticity Analysis of FrcSA

We first examine the monthly lagged correlation of soil moisture and FrcSA with the former preceding the latter. Figure 9A shows the optimal lagged correlation for the contributing catchments for 863 order-1 or order-2 reaches. As shown in Figure 9B, medians of optimal correlations are above 0.4 in all climate divisions except for the southeast desert basins. There is an interesting latitudinal gradient in the correlations which are higher in the northern climate divisions and drop in the central and southern climate divisions (Figure 9B). One possible cause is that the storage-discharge relationship between soil and river water is weaker in a drier climate where soils are not the primary storage of subsurface water. Another explanation can be the higher urbanization and human water use in central and southern California: rivers in the central valley supply irrigational water and the metropolitan areas of dense populations are located along central and southern

coasts with highly regulated rivers. The corresponding optimal lags of FrcSA and soil moisture (in month) is displayed in Figure 9C with the histograms for each climate division in Figure 4D. The majority of catchments show lags of zero month meaning FrcSA and soil moisture largely covary, especially in the three coastal divisions. In mountainous areas and central valleys, more catchments show 2 and 3 months of lag, probably due to greater storage effect from the thicker soil layers.

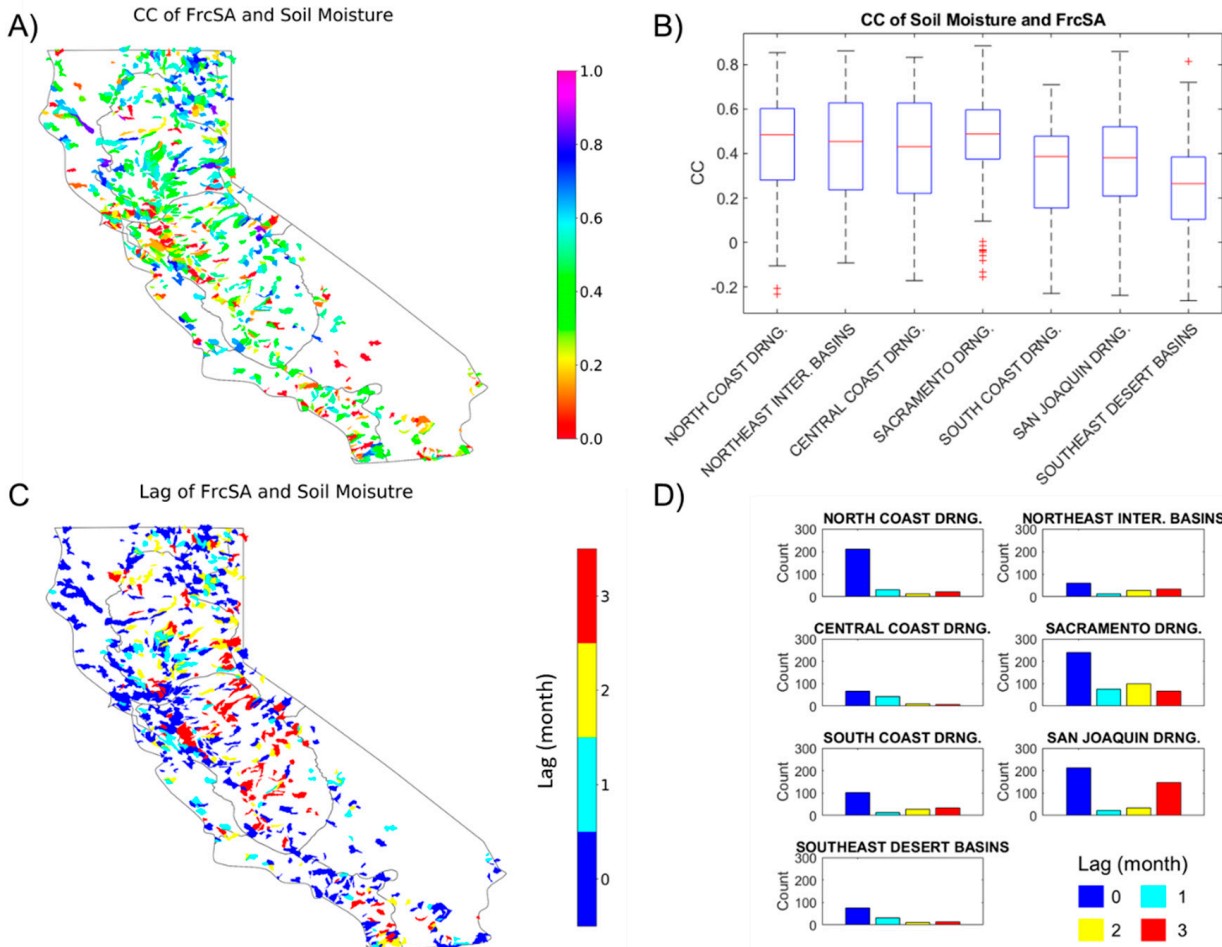

**Figure 9.** (**A**) Optimal lagged correlations of soil moisture and FrcSA for contributing catchments, (**B**) boxplots of the optimal correlations in the seven climate divisions, (**C**) the corresponding optimal lag ranging from 0 to 3 months and (**D**) histogram of optimal lags in the seven climate divisions.

On the yearly scale, we also examine the elasticity of river extent (FrcSA) to cumulative winter rainfall (Pw). Figure 10A shows the times series of normalized FracSA and Pw, which are ratio of annual mean value divided by the long-term mean. These two time-series show good accordance in all climate divisions except for the south coast and southeast desert divisions. In these two exceptional climate divisions, the FrcSA are more stagnant and insensitive to change in precipitation, probably due to the highly regulated nature of the rivers. The elasticity analysis results in 332 reaches with statistically significant results ($p$ value < 0.05). The average elasticity of FrcSA to $P_w$ ($E_{pw}$) is 1.13 (Figure 10B), meaning that a 1% change in the normalized Pw translates to 1.13% of the normalized FrcSA. The $R^2$ value is on average 0.34, meaning the variability of Pw explains 34% of the variability in FrcSA (Figure 10C).

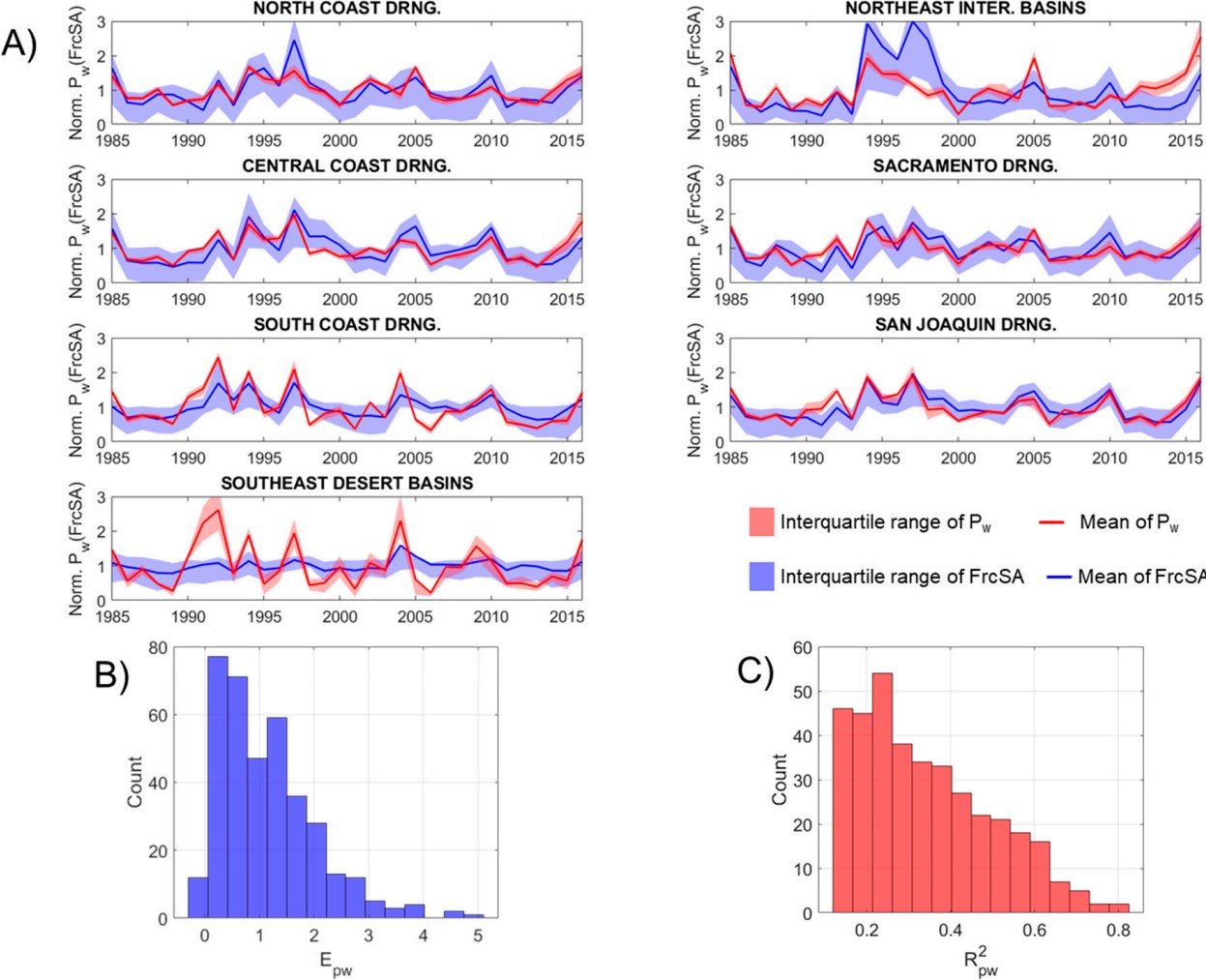

**Figure 10.** Mean and interquartile range of annual mean FrcSA and winter precipitation (Pw) for order-1/order-2 reaches in seven climate divisions (**A**), histograms for the elasticity of FrcSA to $P_w$ ($E_{pw}$) (**B**) for 332 reaches with p value below 0.05 (**B**) and corresponding coefficient of determination ($R^2_{pw}$) (**C**).

We also attribute the variability of $E_{pw}$ into possible causal factors. First, humidity index (P/PET) is overall low in California except for the north coast area and Sierra Nevada (Figure 11A). When examined along with Pw, the increase of humidity index corresponds to the decreasing pattern of Pw (Figure 11B). In the driest catchments (P/PET < 0.2), small increase of P relative to PET is associated with large decrease in Pw. In mountainous Sierra Nevada and north coast (P/PET > 0.40), the $E_{pw}$ is consistently lower and does not vary systematically with P/PET. In addition, the scatters in Figure 11B are colored based on water table depth (WTD). We find that $E_{pw}$ decreases with larger water table depth (WTD) values, indicating catchments with shallow groundwater depth see higher proportional response of FrcSA to variability in winter rainfall. We did not find statistically significant relationships of Pw with other human-induced catchments attributes (not shown), e.g., human water use, degree of flow regulation, or urbanization rate.

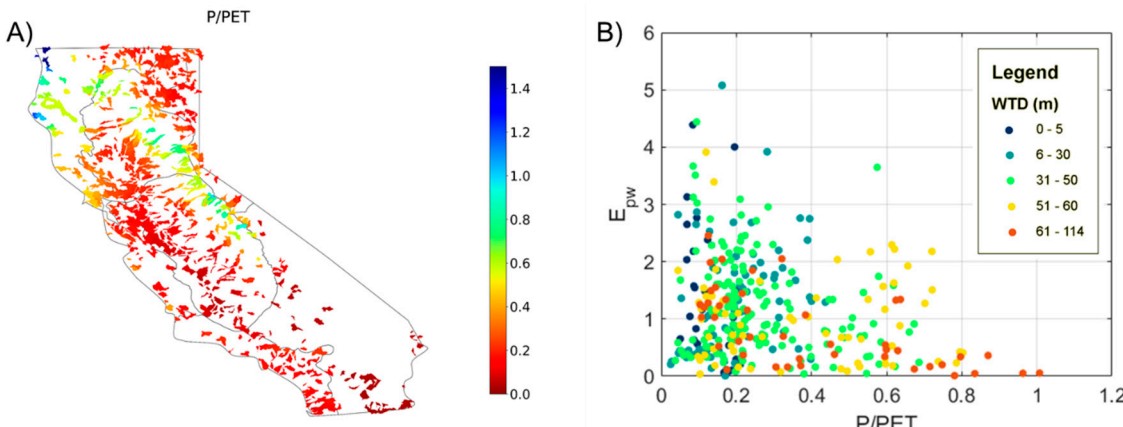

**Figure 11.** Humidity index of order-1/order-2 catchments (**A**), the elasticity of FrcSA to winter precipitation in relation with humidity index and groundwater table depth (WTD) (**B**).

## 4. Discussion

Previous studies on river dynamics rely on the availability of stream gauging network and field surveys. As an indicator of river dynamics, the gauged streamflow estimates are continuous in time, whereas the FrcSA based on Landsat imagery can better represent the spatial variability. Several studies focused on annual low-flow elasticity (extracted from summer streamflow) and explained river behavior via energy demand in relation to moisture supply [49–51]. Similar to our findings, water-limited catchments (i.e., arid and semiarid areas with low P/PET) are more sensitive to climate variabilities than energy-limited catchments (i.e., a humid region with high P/PET). Our results also suggest the important role of subsurface storages in controlling river dynamics: soil moisture is found to correlate with river extent, indicating a storage-discharge relationship (Figure 9). Reager et al. [52] also demonstrated the influence of terrestrial storages on streamflow by correlating GRACE observation with streamflow to infer predictability of river dynamics. In addition, we show that river extent in catchments of deep groundwater depth, particularly in north coast and northern Sierra Nevada, is less sensitive to winter precipitation. Cooper et al. [48] utilized streamflow data and found these mountainous rivers feature slow baseflow recession and reduced elasticity of summer low flow to winter precipitation. In contrast, the shallow and thin aquifers beneath the headwater catchments in northeast interior basins are probably fast-draining and thus more sensitive. The lack of storage in these catchments explains the river extent change before and after the 2012–2015 drought ended (Figure 8): the depleted storage is far below average before the drought ending and slow to recover after the drought ending. Godsey et al. [53] also suggested that the low flow in slow-draining, low-mid-elevation catchments of Sierra Nevada is more resilient against increased evapotranspiration than in fast-draining high-elevation sites. Nonetheless, one should note that elasticities represent change in FrcSA with respect to unit changes in climatic forcing. The absolute change in FrcSA has to be interpreted conditioned on the heterogeneity of climatic forcing. This explains why, in some literature, slow-draining catchments could see larger decline in annl low flow than fast-draining catchments [54,55]. In summary, subsurface water storages, are influential to the *resistance* and *recovery* of river water, as the two components of the *resilience* concepts [56,57].

Although winter precipitation is hypothesized as the first-order control on river extent, we acknowledge that snowfall can be a major component of Pw for some catchments [53,54] and conduct a zoomed-in elasticity analysis of summer mean FrcSA to annual maximum snow water equivalent ($SWE_m$). We select 16 reaches/catchments where the mean ratio of annual cumulative snowfall to precipitation (Snowf/P) is above 30%. Forcing data from NLDAS are used to calculate Snowf/P as shown in Figure 12A. At a yearly scale, the time series of annual maximum SWE and summer mean FrcSA from the 16 reaches show certain level of accordance (Figure 12B). The resulted elasticity (Eswe) values are on average 0.4

(Figure 12B) with p values all below 0.05 and the mean $R^2$ is 0.44 (Figure 12C). Therefore, as a small portion of all (332) examined reaches, the snow-dominant reaches do not show exclusively different elasticity of summer mean FrcSA to $SWE_m$, as compared to $E_{pw}$.

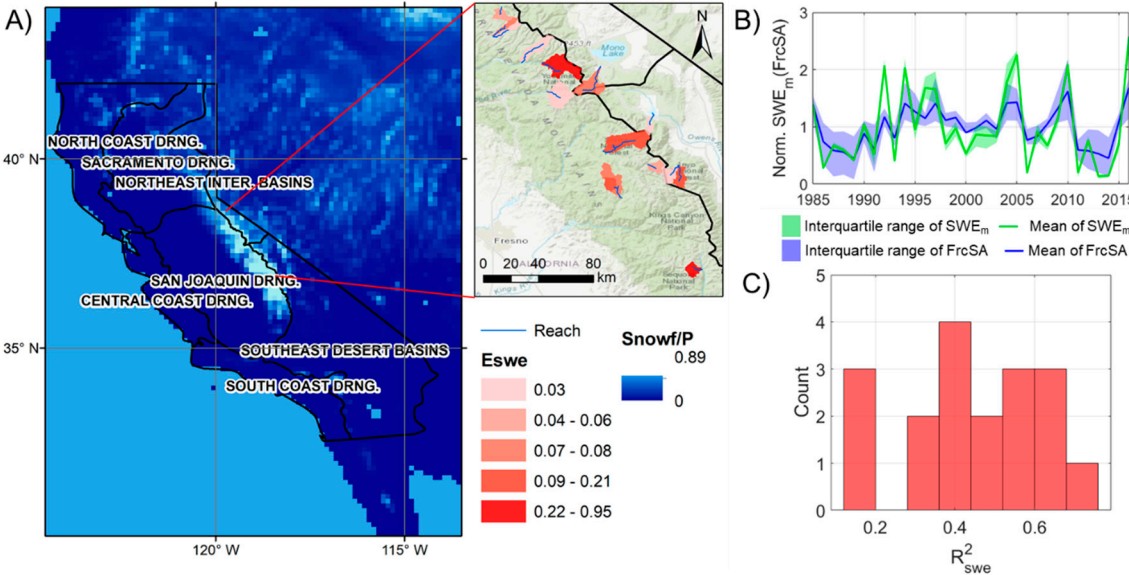

**Figure 12.** Mean annual fraction of snowfall in precipitation (Snow/P) in the study region and elasticity of FrcSA to maximum annual snow water equivalent ($E_{swe}$) in 16 snow-dominant catchments (Snow/P >= 0.25) (**A**), Mean and interquartile range of normalized annual mean FrcSA and annual maximum SWE ($SWE_m$) for the 16 catchments (**B**) and the corresponding coefficient of determination ($R^2_{pw}$) for the elasticity analysis (**C**).

The long record of the Landsat archive makes it optimal for analyzing multiple historical droughts, but there are inherent limitations of the dataset bringing uncertainties into the river extent estimation. First, GSW are of 30-m resolution, which can be too coarse for detecting river extent during the lowest flow conditions. Figure 13A shows the probability distribution of monthly mean streamflow from the 187 USGS gauges when the co-located reaches show zero FrcSA. The majority of such cases occur during droughts with the median PDSI being around-2 (Figure 13B). Out of the total 2060 FrcSA values, over half (1060) correspond to the streamflow lower than 0.1 cm, which is the no-flow threshold found by Zimmer et al. [58]. The remaining cases with higher streamflow represent the uncertainty from the Landsat observation: the estimated river extent is subject to negative bias because the surface water body smaller than the 30-m scale may not be detected. Therefore, in Figure 6, the increase in perennial/intermittent area and decrease in dry area after the drought ending can be slightly underestimated. In other words, the change of FrcSA across the drought ending can be more significant than what is detected by Landsat. As a remedy to the 30-m resolution, Jones et al. [59] developed algorithms to improve water detectability of Landsat at the subpixel level and produced the dynamic surface water extent (DSWE) dataset. Walker et al. [60] reconstructed the monthly history of surface water based on DSWE for central valley region in California and the comparison with GSW indicates strong agreement between the two datasets. Another limitation of the Landsat data is clearly the poor temporal coverage during winter months (Figure 3), which prevents us from fully investigating the intra-annual variability, i.e., the wetting period of rivers. The increase of river water across drought endings would be even more drastic than estimated in this study, if good Landsat observations could be achieved during winter months when most drought-ending storms occur (Table S1). As an alternative to optical sensors, active sensors such as synthetic aperture radars (SAR) have advantages in terms of spatial and temporal variation of surface water, including 'all-weather' and 'day-and-night' capacities, as well as sensitivity to open water and below-canopy inundation [61–64]. Therefore, future studies on river drying can employ SAR imagery for mapping surface water extent.

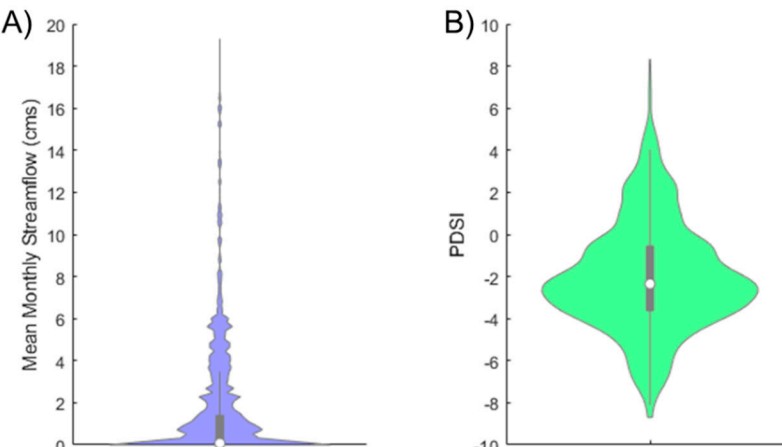

**Figure 13.** Violin plots (probability distribution) of mean monthly streamflow (**A**) and PDSI (**B**) when FrcSA is zero at co-located reaches. The sample size for the plots is 2060.

## 5. Conclusions

In this study, we investigate the dynamics of river extent in California using Landsat observation, focusing on during- and post-drought periods. We used river centerline vectors and GSW monthly history observation in combination to achieve monthly fractional river extent (FrcSA) for 2071 river reaches. As an indirect comparison, we compare the FrcSA with monthly mean streamflow from USGS gauges. We also examine how Landsat datasets characterize river dynamics during climatic extremes using the 2012–2015 drought as a typical event. In order to attribute river dynamics to natural causal factors, we associate the FrcSA time series with forcings and model states from the North American Land Data Assimilation System (NLDAS) via correlation and elasticity analyses. The study is among the first to utilize Landsat datasets for analyzing spatiotemporal variability of river extent in an arid and semi-arid environment. The following findings emerge from our analysis:

1.  The Landsat provides consistent observations of over 90% of the river area within bankful widths from March to October, but poor observations from November to February, making it a viable source for monitoring rivers' seasonal drying dynamics (from spring to fall). When compared with streamflow at co-located USGS gauges, FrcSA shows fair (>0.5) correlation except for in southern dry climate divisions and high-elevation mountainous locations. The 30-m resolution causes potential underestimation of FrcSA during low flow conditions.

2.  During the 2012 to 2015 California drought and the subsequent 2016 and 2017 wet years, the rivers in California experienced the two extremes (dry and wet) of regional climatology, which is well reflected by the Landsat observation. The river extents exhibit drastic swings before and after the drought ending. River extent in water-scarce areas is more resilient to droughts because of the high degree of regulation. Rivers in northeast interior basins are significantly dried during droughts and slow to recover after drought endings due to small catchment storage.

3.  At a monthly scale, soil water, as a component of terrestrial water storage, shows a latitudinal gradient in the correlation with FrcSA: river extent in the humid catchments is better correlated with soil moisture than in the arid southern region. Longer lags of FrcSA following soil moisture are seen in Sierra Nevada than in coastal regions, due to the larger storage of thick soil layers. At a yearly scale, elasticity of summer-mean-FrcSA to winter precipitation ($E_{pw}$) is statistically significant for over 300 low-order (1 or 2) reaches. Catchments with shallow groundwater table show high elasticity in FrcSA to Pw because their small storge favors fast draining of baseflow. Both correlation and elasticity analyses validate the hypothesis that a storage-discharge relationship controls the dynamics of river extent.

This study provides insight into the range and impacts of drought on flow regime, which broadens and augments previous studies based on stream gauge records. In general, the river extent during the seasonal drying (from spring to fall) in California is driven by precipitation as the first-order control. Given longer droughts in a future climate, fast-draining rivers controlled by surface runoff are likely to experience the largest-percent declines [48]. The reduced precipitation, coupled with additional stressors such as increased extreme event frequency and higher temperatures, will be critical for predicting the effects of climate change on stream ecosystems and human–ecosystem interactions in the 21st century [65,66]. Future climate projections have raised alarms that the expected changes will be great enough to exceed boundaries of ecosystem resilience [67–69] and damage ecosystem services [70–72].

**Supplementary Materials:** The following are available online at https://www.mdpi.com/article/10.3390/rs13173423/s1, Figure S1: Monthly PDSI time series in the seven climate divisions with the identified drought periods (red shade) using threshold of PDSI = −2 (red dotted line), Table S1: Summary of the drought events identified based on PDSI time series in the seven climate divisions in California, Table S2: Summary of abbreviations.

**Author Contributions:** Conceptualization, S.G. and Y.H.; methodology, S.G.; software, S.G.; valida-tion, S.G. and Z.L.; formal analysis, S.G.; investigation, S.G., Z.L. and M.C.; resources, S.G., D.A., T.N. and Z.L.; data curation, S.G. and Z.L.; writing—original draft preparation, S.G.; writing—review and editing, S.G., Z.L., M.C., D.A., T.N. and Y.H.; visualization, S.G.; supervision, D.A., T.N. and Y.H.; project administration, D.A., T.N. and Y.H.; funding acquisition, D.A., T.N. and Y.H. All authors have read and agreed to the published version of the manuscript.

**Funding:** This study is supported by the National Science Foundation (Project number: 1802872).

**Data Availability Statement:** Not applicable.

**Acknowledgments:** The authors would also like to thank the funding support from National Science Foundation (Project number: 1802872).

**Conflicts of Interest:** The authors declare no conflict of interest.

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
