# Peer review of "Monitoring Drought through the Lens of Landsat: Drying of Rivers during the California Droughts"

_remotesensing, doi:10.3390/rs13173423_

Round 1
Reviewer 1 Report
Reviewer’s Report on the manuscript entitled:
Monitoring Drought through the Lens of Landsat: Drying of Rivers During the California Droughts
The authors investigated the river extent dynamics in California via Landsat imagery for during and post-drought periods. They also examined the fractional River Extend (FrcSA) in 2071 river reaches and its correlation with streamflow at co-located USGS gauges. Using some statistical analyses, they also related FrcSA with hydrometric variables.
The paper is generally well-written and suitable for publication in Remote Sensing but the format, figures, locations, page numbers, line numbers are either missing or incorrectly inserted. In addition, I have several comments for authors to improve their paper:
1) As I mentioned, unfortunately, the page numbers were inserted incorrectly. All show “3 of 4” at the top of each page for some reason. Furthermore, the line numbers are missing! When you prepare the revision, please carefully check the Figure numbers, Table numbers, Page numbers, and please add the line numbers before resubmission.
2) The labels in the figures should be enlarged a bit, and the figure quality should be enhanced.
3) Page 3. Line 1. Please use a hyphen for “inter-basin” instead of a dot.
4) Water table Depth (WTD) appears in the caption of Figure 1 for the first time. So, the acronym should be defined there. The second time it appeared is on page 12.
5) Section 2.2.1 Please define the acronym NOAA. (National Oceanic and Atmospheric Administration) although it is known to many.
6) Section 2.2. please move the links to the References according to MDPI guidelines. So please move ALL the links to the references and simply cite in in the body of the manuscript just like other references.
7) Section 3.1. That should be Figure 2. It is currently labeled as Figure 1.
8) Figure 3. Format issue. The name Figure is italic for some reason. Same for Figure 4, Table 1,
9) In Figure 3, the labels must be enlarged. They are currently tiny and almost unreadable. Furthermore, the FracSA has data gaps, but the authors made no comments on how they treat the gaps in the manuscript. Please clarify.
10) Figure 7 on page 9. Why the period 2009 -2015 is excluded from the box plot graphs? Please clarify.
11) The figure shown on page 10 can be improved and the four panels can be enlarged with higher quality.
12) The discussion part should be further improved in light of other studies. For example, I suggest authors read the following articles and include them there:
As the authors mention, the limitation of Landsat imagery in months November to February, other studies https://doi.org/10.3390/rs10050797 and https://doi.org/10.1080/01431161003749477 have used Sentinel 1 SAR images that are more frequently acquired with finer spatial resolution and not impacted by clouds and sunlight, though they have their own limitations (e.g., challenges to separate land from the water). A combination of Sentinel 1 SAR imagery and Landsat imagery along with discharge and climate data sets could be a robust approach for monitoring river networks.
A cross-wavelet analysis method in https://doi.org/10.3390/rs12152446 is used to decompose multiple time series into the time-frequency domain to estimate the coherency and phase differences (e.g., for time lag estimation in Section 3.2) between the time series components, e.g., Landsat image and climate time series. Furthermore, a monitoring technique for disturbance detection (e.g., drought) is also proposed for unevenly sampled Landsat image time series with gaps. This could be described in the light of the methodology used in the paper.
13) An Abbreviation Table should be added at the end of the manuscript to list all the acronyms used in the manuscript.
14) Adding a flowchart describing the data processing methods used in the manuscript can help readers to better understand the methodology.
Regards,
Reviewer 2 Report
This is an interesting work and such type of work is albeit ferwe in the literature. Hence this would can potentially enhance the connection between droughts and drying up of rivers. However, there are few issues that should be accounted for to improve the readership of this work as it currently stands. I have outlined my observations below:
[1] Literature definitively requires works from other parts of the world especially from humid region where surface waters are highly impacted by droughts. This can be improved by considering the following works
https://www.sciencedirect.com/science/article/pii/S0169809520310917
https://link.springer.com/article/10.1007/s00477-020-01768-2
https://www.sciencedirect.com/science/article/pii/S0048969720356886
Three objectives have been used but results are not clearly explained for objective 2 and partial for objective 1.
[2] p. 3 – ‘… for the seven climate divisions ..’ – is this in study area? For international readers, you are to clarify this statement where you are stating this for the first time. Should add ‘the’ in the statement on this page ‘’We identify droughts in the same..”
[3] Is there any evidence resampling of 1/8° to 1/160°? I am really curious to see this
[4] in section 3.1 you are saying ‘wetting of rivers in winter months’ – I’d question why? Clarification is needed. Fig. 1 does not say the story if someone is not reading the texts. Hence this figure caption must be clarified
[5] Results section is too large with too many figures. I suggest keeping only interesting results and the rest can be sent to sup info
[6] Discussion section essentially is the poorest and too short. Hence this section must be revamping and developed accordingly. In the discussion section, you have the opportunity to relate your findings with theory, if there are any similarities or dissimilarities you can argue causes of similarity and dissimilarities by drawing examples from other works such as those listed above
[7] Conclusion section is large and does not reflect aims at hand. Hence your attention is needed.
Round 2
Reviewer 1 Report
I would like to thank the authors for addressing most of my comments. The manuscript and qualities of figures are improved. However, my comment #12 regarding the references and discussion is partially addressed:
Please also add the following two references in the last paragraph of the Discussion (Section 4):
1) https://doi.org/10.1080/01431161003749477
2) https://doi.org/10.3390/rs12152446
The first reference above proposes a river detection algorithm that has used Sentinel 1 SAR images that are more frequently acquired with finer spatial resolution and not impacted by clouds and sunlight, though they have their own limitations (e.g., challenges to separate land from the water in windy days).
The second reference above proposes a monitoring technique for change detection within unevenly sampled satellite image time series with gaps (e.g., data gaps in winter). The method can be applied to per-pixel time series for monitoring rivers that may show seasonal patterns. The cross-wavelet analysis method in this article can also be used for estimating coherency and time lag (e.g., in Section 3.2, coherency and phase lag between FrcSA and hydroclimate variables).
Authors may also mention the articles above for future work in the manuscript.
Please carefully proofread the article and to ensure that figures and references to figures are correct.
Thank you for your contribution
Regards,
Reviewer 2 Report
no comment